# Saccade-synchronized rapid attention shifts in macaque visual cortical area MT

Tao Yao [1,2], Stefan Treue[1,3,4,5] & B. Suresh Krishna [1,4]

While making saccadic eye-movements to scan a visual scene, humans and monkeys are able to keep track of relevant visual stimuli by maintaining spatial attention on them. This ability requires a shift of attentional modulation from the neuronal population representing the relevant stimulus pre-saccadically to the one representing it post-saccadically. For optimal performance, this trans-saccadic attention shift should be rapid and saccade-synchronized. Whether this is so is not known. We trained two rhesus monkeys to make saccades while maintaining covert attention at a fixed spatial location. We show that the trans-saccadic attention shift in cortical visual medial temporal (MT) area is well synchronized to saccades. Attentional modulation crosses over from the pre-saccadic to the post-saccadic neuronal representation by about 50 ms after a saccade. Taking response latency into account, the trans-saccadic attention shift is well timed to maintain spatial attention on relevant stimuli, so that they can be optimally tracked and processed across saccades.

[1] Cognitive Neuroscience Laboratory, German Primate Center–Leibniz Institute for Primate Research, 37077 Goettingen, Germany. [2] Laboratory for Neuro- and Psychophysiology, KU Leuven Medical School, Campus Gasthuisberg, 3000 Leuven, Belgium. [3] Bernstein Center for Computational Neuroscience, 37077 Goettingen, Germany. [4] Leibniz-ScienceCampus Primate Cognition, 37077 Goettingen, Germany. [5] Faculty of Biology and Psychology, University of Goettingen, 37073 Goettingen, Germany. Correspondence and requests for materials should be addressed to B.S.K. (email: skrishna@dpz.eu)

Humans and monkeys are able to keep track of relevant visual stimuli while making saccadic eye movements to scan a visual scene. Since the visual system mostly operates using retinotopic representations[1–3], in each visual area, a relevant visual stimulus (the target) at a fixed spatial location is represented by one neuronal population before the saccade and a different neuronal population after the saccade: we refer to these as the pre-saccadic and post-saccadic target representation, respectively. As a result, to maximally and selectively enhance target processing both before and after the saccade, a rapid, saccade-synchronized shift of spatial attentional modulation from the pre-saccadic to the post-saccadic target representation is optimal. Enhancement by spatial attention would ideally be expected to be dominant at the pre-saccadic target representation until just before the saccade begins, and decay at or soon after the saccade ends. Similarly, attentional enhancement would be expected to emerge at the post-saccadic target representation at or soon after the saccade ends. In other words, if attentional enhancement of the pre-saccadic target representation decayed well before the saccade, or attentional enhancement of the post-saccadic target representation emerged well after the saccade, there would be time-periods where the target stimulus did not receive the benefits of attentional allocation. Contrariwise, if attentional enhancement of the pre-saccadic target representation lingered after the saccade, or attentional enhancement of the post-saccadic target representation emerged pre-emptively well before the saccade, attention would be peri-saccadically allocated to irrelevant spatial locations, distractor processing would potentially be facilitated and this would degrade task performance.

Until now, to our knowledge, the time-course of the shift of attentional modulation from the pre-saccadic to the post-saccadic target representation across a saccade has not been explicitly measured. In the only previous physiological recording study on this issue, using a saccade task similar to ours with a fixed attentional target, object-based attentional enhancement of multiunit activity in monkey V1 was reported to emerge in the post-saccadic target representation approximately 80 ms after the end of the saccade[4]. However, this study did not measure the dynamics of the decay of attentional enhancement in the pre-saccadic target representation. On the other hand, in a human imaging study, functional magnetic resonance imaging (fMRI) and electroencephalogram (EEG) data from humans have been taken as evidence for attentional modulation lingering for about 100 ms after the saccade in the pre-saccadic target representation;[5] an interpretation supported by results from human psychophysical studies[6,7]. Human psychophysical data consistent with an early, pre-saccadic emergence of attentional modulation in the post-saccadic target representation have also been reported[8,9]. This psychophysical inference of pre-emptive attentional modulation in the post-saccadic target representation is consistent with a large body of single-neuron recording data from putative attentional control regions in monkeys showing that neurons in the lateral intraparietal area, superior colliculus and frontal eye field[3,10–12] respond predictively when a stimulus was expected in their receptive field (RF) after the saccade. This predictive activity is greater in the lateral intraparietal area (LIP) for stimuli with greater bottom-up saliency[13] and for stimuli that are learnt visual search targets[14] or saccade targets[13].

Though these results are suggestive (see Discussion), they do not address the time-course over which attentional enhancement is remapped from the pre-saccadic to the post-saccadic target representation across a saccade. In order to measure the time-course of this trans-saccadic attention shift, we trained two monkeys to make saccades while maintaining attention on a moving random dot pattern (RDP) at a fixed spatial location. We recorded from visual area MT, a key locus in the cortical motion-processing pathway of humans and monkeys, where neurons show both small RFs and clear, robust, attentional enhancement[15–18]. We show for the first time that the trans-saccadic shift of attentional enhancement is well synchronized to saccades and that attentional enhancement crosses over from the pre-saccadic to the post-saccadic target stimulus representation soon after saccade offset. We recently showed that in humans performing a similar task, spatial attention is fully available at the task-relevant location within 30 ms after the saccade[19]. Taking visual response latency into account (see Discussion), this rapid post-saccadic availability of spatial attention at the task-relevant location in humans is in excellent agreement with the physiological time-course of trans-saccadic attention shifts in monkeys that we report here. Our results show that the trans-saccadic attention shift in primates is precisely co-ordinated with the saccade to maintain attentional enhancement on relevant stimuli, so that they can be attentionally enhanced soon after the beginning of each eye fixation and can thus be optimally tracked and processed across saccades.

## Results

**Peri-saccadic attentional task**. Our experimental paradigm required monkeys to maintain attention on one of four moving RDPs while also making a saccade; we refer to this attended RDP as the target and the other three as distractors. We recorded from neurons in area MT during this task. We carefully chose the locations of the four RDPs so that a target or distractor RDP lay in the RF of the recorded neuron before or after the saccade. Since MT neurons have retinotopic RFs whose spatial location moves with each saccade (Fig. 1a), the attended target RDP lay in the RF of (and was therefore represented by) different populations of neurons before and after the saccade. In Experiment 1, we estimated the attentional enhancement of the pre-saccadic target representation, while in Experiment 2, we estimated the attentional enhancement of the post-saccadic target representation. To do this, in Experiment 1, we placed the attended RDP so that before the saccade, it lay either in the RF (the attend-in condition) or meridionally opposite to it (the attend-out condition): we measured the attentional enhancement of the pre-saccadic target representation by comparing the firing-rates in the attend-in and attend-out conditions (upper part of Fig. 1b, c). In contrast, in Experiment 2, we placed the attended RDP so that after the saccade, it lay either in the RF (the attend-in condition) or meridionally opposite to it (the attend-out condition): we now measured the attentional enhancement of the post-saccadic target representation by comparing the firing-rates in the attend-in and attend-out conditions (lower part of Fig. 1b, d).

**Peri-saccadic attentional response dynamics**. Based on prior findings[17,20], we expected to see an attentional enhancement of the pre-saccadic target representation (in Experiment 1) before the saccade and of the post-saccadic target representation (in Experiment 2) after the saccade. This is indeed what we found. In Experiment 1, the population average PSTH shows a greater response before the saccade when a target RDP, rather than a distractor RDP, appeared in the neuron's RF before the saccade (attend-in condition: cyan vs. attend-out condition: magenta curves in Fig. 1c); the stimulus in the RF after the saccade was always a distractor. In Experiment 2, the population average PSTH shows a greater response after the saccade when a target RDP, rather than a distractor RDP, appeared in the neuron's RF after the saccade (attend-in condition: blue vs. attend-out condition: red curves in Fig. 1d); the stimulus in the RF before the saccade was always a distractor. We defined and estimated the

attentional enhancement as the difference in firing-rates between attend-in and attend-out conditions. For the pre-saccadic target representation in Experiment 1, there is significant attentional enhancement in the time-window from 0–500 ms before saccade onset (Monkey H: $5.2 \pm 0.9$ Hz, 11.3% enhancement, $p < 0.0001$, paired $t$-test; Monkey E: $6.0 \pm 1.0$ Hz, 31.3% enhancement, $p < 0.0001$), but not from 0–500 ms after saccade offset (monkey H: $p = 0.16$, monkey E: $p = 0.06$). For the post-saccadic target representation in Experiment 2, there is significant attentional enhancement in the time-window from 0–500 ms after saccade

offset (Monkey H: $6.0 \pm 0.9$ Hz, 19.2% enhancement, $p < 0.0001$; Monkey E: $6.1 \pm 1.0$ Hz, 19.9% enhancement, $p < 0.0001$), but not from 0–500 ms before saccade onset (monkey H: $p = 0.16$, monkey E: $p = 0.51$). Consistent with these results from the separate significance tests, a direct statistical comparison between the attentional effects in the two time-periods also shows that the attentional effect in the pre-saccadic period is greater than that in the post-saccadic period in Experiment 1 (paired $t$-test; $p < 0.0001$ in both monkeys), and the attentional effect in the post-saccadic period is greater than that in the pre-saccadic period in

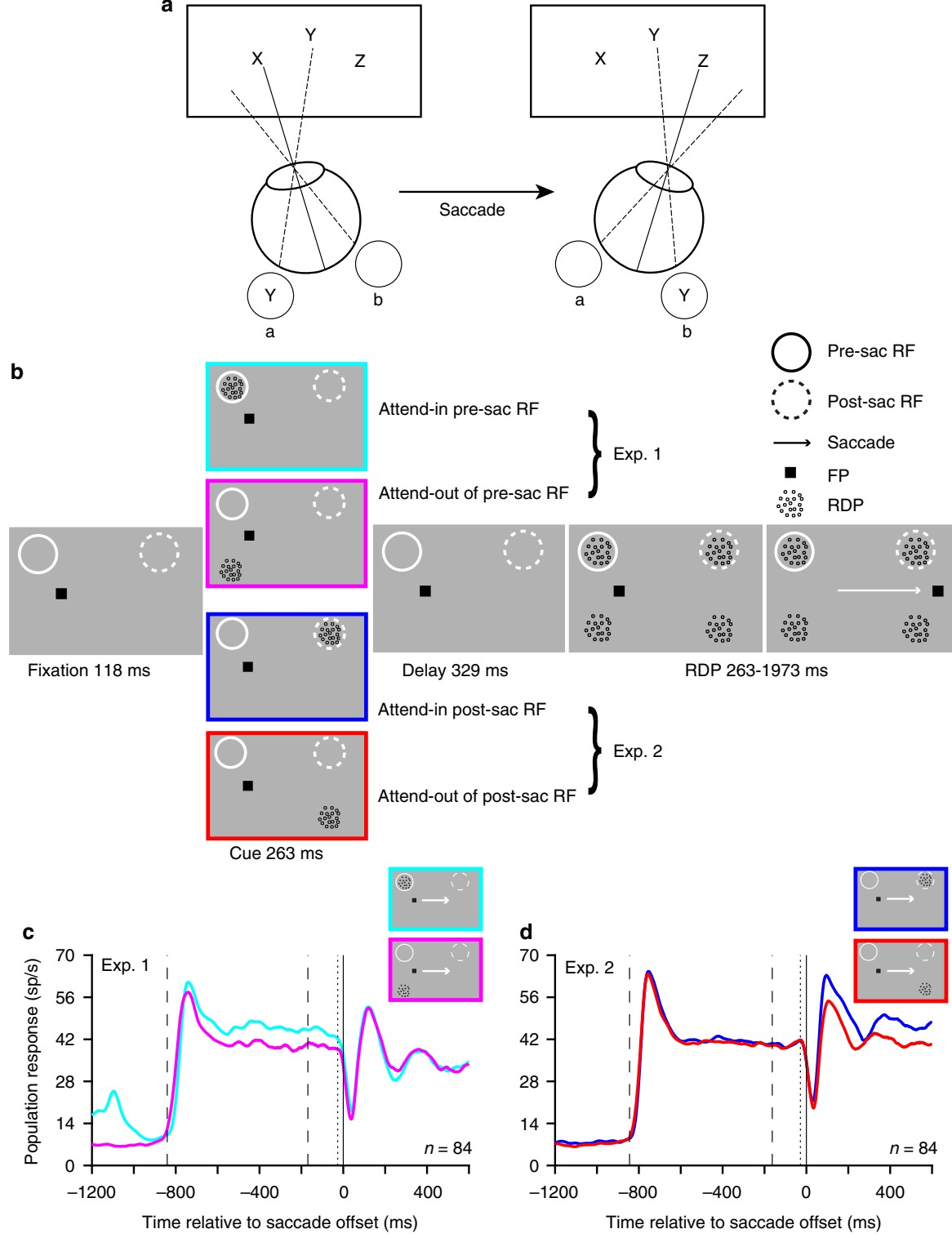

Experiment 2 (paired *t*-test; monkey H: $p < 0.0001$, monkey E: $p = 0.001$).

These results confirm, as expected, that an attentional response enhancement is found in the pre-saccadic target representation before the saccade, and in the post-saccadic target representation after the saccade. Our primary goal was to characterize the time-course of this shift of attentional enhancement from the pre-saccadic to the post-saccadic target representation and measure how well this shift was synchronized to the saccade. To do this, we focused on the time-interval from 200 ms before to 200 ms after saccade offset during which the attention shift takes place (Fig. 2). For best task performance, the attention shift should happen as close to saccade offset as possible so that the pre-saccadic target representation is enhanced right up until saccade offset and the post-saccadic target representation soon after saccade offset. We find that attentional enhancement of the pre-saccadic target representation (Experiment 1) is statistically significant throughout the period before saccade offset ($-200$ ms to 0 ms before saccade offset) and until 100 ms after saccade offset in monkey H and 50 ms in monkey E (gray curves in Fig. 2a, b; significance based on 50 ms non-overlapping time-bins). The attentional enhancement of the post-saccadic target representation (Experiment 2) becomes statistically significant right after saccade offset in monkey H and after 50 ms of saccade offset in monkey E (black curves in Fig. 2a, b). Comparing the two representations directly, the attentional effect in the post-saccadic target representation becomes larger than the attentional effect in the pre-saccadic target representation at 29 ms (monkey H) and 53 ms (monkey E) after saccade offset; we call this time the attentional cross-over time. To estimate the variability of this cross-over time, we used a bootstrap procedure to calculate an inter-quartile range (IQR: see Methods): the IQRs were 10 ms (monkey H) and 11 ms (monkey E). The cross-over times calculated using a 15-ms standard-deviation Gaussian-filtered PSTH are very close to those obtained using a smaller standard deviation of 10 ms (monkey H: 28 ms and monkey E: 54 ms). The proximity of the attentional cross-over to saccade offset indicates that the attention shift is well synchronized to the saccade, after taking the visual response latency of MT neurons into consideration (see Discussion). Importantly, we did not find any evidence for predictive attention shifts in MT: there is no attentional enhancement of the post-saccadic target representation before saccade offset (black diamonds in Fig. 2a, b). This is particularly notable because unlike earlier studies[18,21], we made sure that there was a stimulus in the RF before the saccade. The presence of this stimulus ensured a stimulus-driven response on which a putative predictive attentional signal could act, and rules out the argument that the apparent absence of a predictive response is simply because the predictive attentional signal does not modulate spontaneous activity in MT. Finally, in a series of additional control analyses, our conclusions about the rapid post-saccadic dynamics of attention shift remain robust when matching for firing-rate across neurons in Experiments 1 and 2 (Supplementary Fig. 1), when examined using a ratio measure (Supplementary Fig. 2), when examined in the neurons recorded in both tasks in monkey H (Supplementary Fig. 3), when matching the saccade offset-time distributions between attend-in and attend-out conditions for each neuron (Supplementary Fig. 4), and when imposing a more stringent control on the post-saccadic eye position (Supplementary Fig. 5).

**Peri-saccadic attentional shifts are saccade-synchronized**. The results presented above indicate that the trans-saccadic attention shift is well synchronized with saccade planning/execution. We examined the alternative possibility that the attention shift and saccade planning/execution were both triggered by the cue to make the saccade, but proceeded independently. If the attention shift and saccade planning/execution proceed independently, then relative to saccade offset, the time-course of the attention shift would be delayed for trials with short-latency saccades compared to trials with long-latency saccades. On the other hand, if the attention shift and saccade planning/execution are indeed synchronized with each other, then the time-course of the attention shift would be similar for short and long-latency saccades. At the same time, in this scenario, relative to the cue to make the saccade (i.e., the offset of the fixation point), the attention shift would occur later on trials with long-latency saccades compared to short-latency saccades (at times after the mean saccade latency, i.e., once the saccade had occurred). We therefore plotted the time-course of the attention shift aligned to saccade offset (Fig. 3a, c) and fixation point offset (Fig. 3b, d) for trials with saccade latencies shorter than the median latency and longer than the median latency (calculated for each recording session; red and blue curves respectively; also see Supplementary Figs 6 and 7). We also matched the latency distributions for the attend-in and attend-out conditions for each neuron for these analyses to avoid any potential confounds due to systematic latency variations between the two conditions (also see Supplementary Note 1). When aligned to saccade offset (Fig. 3), the attention shift's time-course for trials with long-latency saccades and trials with short-latency saccades is generally superimposed, consistent with the pattern expected if the attention shift was synchronized with saccade planning/execution. Also, as expected from this pattern, when aligned to fixation point offset (i.e., the cue to make the saccade), the attention shift's time-course for trials with long-latency saccades is delayed (compared to the trials with short-latency saccades) at times after the mean saccade latency. Though

**Fig. 1** Attention enhances different target populations before and after a saccade. **a** Cartoon demonstrating that different neurons (retinotopically) represent an attended stimulus across a saccade. Stimulus X is foveated before the saccade and stimulus Z after the saccade. The attended stimulus Y falls in the RF of neurons representing retinal location a before the saccade and in the RF of neurons representing retinal location b after the saccade. **b** Two rhesus monkeys were trained to covertly attend to one of four moving RDPs (the target) while also making a visually guided saccade if the fixation point (FP) jumped to a new location (after 673 ms). An initial spatial cue marked the target location on each trial. A different RDP appeared in the RF before and after the saccade. In Experiment 1, after the saccade, the stimulus in the RF was always a distractor, while before the saccade, either a target or a distractor appeared in the neuron's RF; the situation was reversed for Experiment 2. The target change occurred between 263 and 1973 ms after RDP onset. **c** Population average peri-stimulus time histograms (PSTHs) from 84 neurons in Experiment 1, aligned to saccade offset. Before the saccade, neurons respond more strongly to a target stimulus (cyan curve) compared to a distractor stimulus (magenta curve). PSTHs constructed by filtering the spike-trains with a truncated Gaussian window (15 ms standard deviation, 100 ms filter width) stepping every 1 ms (see Methods). Inset rectangles: cue location for the two conditions. Left to right: first dashed vertical line—mean time of RDP onset, second dashed vertical line—mean time of fixation point jump, dotted vertical line—mean time of saccade onset. The early response before RDP onset in the attend-in condition (cyan curve) is the cue response. Data are pooled from both monkeys. **d** PSTHs for Experiment 2; 84 neurons. After the saccade, neurons respond more strongly to a target stimulus (blue curve) compared to a distractor stimulus (red curve)

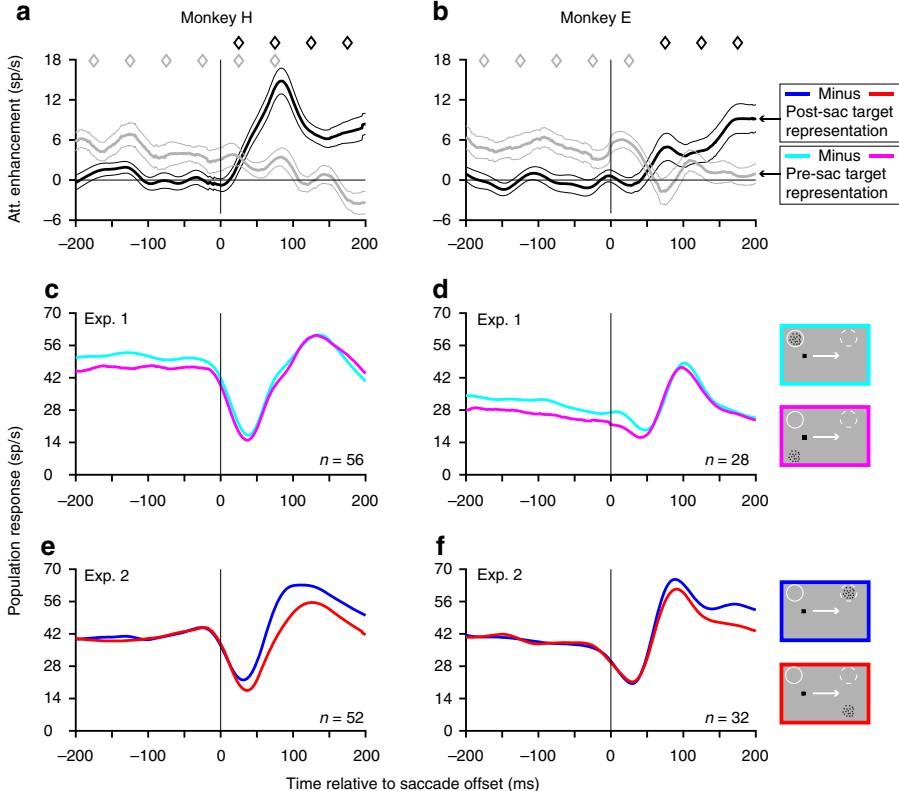

**Fig. 2** The trans-saccadic attention shift is well synchronized to the saccade. **a**, **b** The attentional cross-over time (when the attentional enhancement of the post-saccadic target representation becomes larger than the attentional enhancement of the pre-saccadic target representation) occurs at 29 ms (**a**, monkey H) and 53 ms (**b**, monkey E) after saccade offset. Data are for monkey H in **a** and for monkey E in **b**. Gray and black curves show the mean difference (and s.e.m) between the target-in-RF and distractor-in-RF curves in Experiment 1 and Experiment 2, respectively (as shown in **c**–**f**). Diamonds above the curves indicate the successive, non-overlapping 50 ms time-bins in which the differences are significantly larger than zero (one-sided $t$-test): black diamonds for the black curve and gray diamonds for the gray curve. **c**–**f** Same data as in Fig. 1b (**c**, **d**) and Fig. 1c (**e**, **f**), but plotted separately for the two monkeys and focusing on the time around the saccade (−200 to 200 ms relative to saccade offset); format otherwise identical. The gray curves in **a** and **b** are computed as the difference between the cyan and magenta curves in **c** and **d**, respectively, while the black curves in **a** and **b** are computed as the difference between the blue and red curves in **e** and **f**, respectively. Also see Supplementary Fig. 1

the patterns are noisier, the conclusion is also supported by the data when plotted for each monkey separately (Supplementary Fig. 7).

## Discussion

We report, for the first time, that trans-saccadic attention shifts are well synchronized to saccades: attentional enhancement crosses over from the pre-saccadic to the post-saccadic target representation at 31 and 52 ms after saccade offset in the two monkeys (Fig. 2). We recently reported results from humans performing a task that was very similar to the present study, where they had to respond to a brief 23 ms target motion change and ignore equally brief distractor motion changes[19]. There, human subjects were able to detect target motion changes (and ignore distractor motion changes) occurring 30 ms after saccade offset as well as they did for changes occurring at later times after the saccade. This indicated that the neural response to a change occurring only 30 ms after saccade offset receives sufficient attentional enhancement that the detection performance for this change is as good as that for changes occurring at later times. Our physiological data here provide critical supporting evidence for this inference. Given an onset latency of approximately 40 ms in MT[22] (42.5 ms in our data—see below and Fig. 1), a visual change occurring 30 ms after saccade offset would reach MT by 70 ms after saccade offset. We show here that by this time, attentional

enhancement would have crossed-over to the post-saccadic target representation, thus enabling the change to receive the benefits of attentional enhancement. We note that though attention crosses over to the post-saccadic target representation within 50 ms of saccade offset, attentional enhancement of the post-saccadic target representation does not reach its peak value by that time (Fig. 2a, b). It is possible that for the task conditions used in Yao et al.[19], a level of attentional enhancement sufficient to allow peak performance was already reached by 70 ms after the saccade. The use of a brief 23 ms change in the human study was critical to allow temporally fine-grained measurements of attentional performance to be made. With the longer-duration motion changes of 132 ms that we used in the present study (that are much longer than the typical saccade duration), we did not observe any effect of the saccade on detection performance (Supplementary Fig. 8). We were therefore not able to compare the behavioral performance meaningfully to the dynamics of the trans-saccadic attention shift. We do note that the reduction in detection performance around the saccade cue onset, that was visible in the human behavioral data, is not visible in the MT activity we record here. One possibility is that the observed reduction in performance around saccade cue onset is mediated by areas other than MT. This would be consistent with a recent study showing that inactivating the superior colliculus can lead to large behavioral deficits in a very similar task to ours, without affecting MT and MST neural activity[23]. More precise studies of the relationship

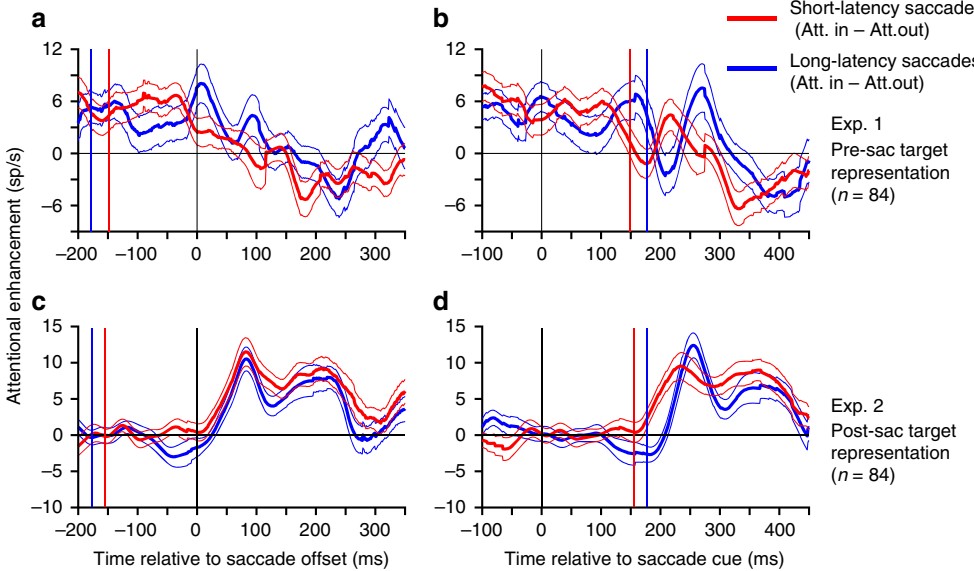

**Fig. 3** The trans-saccadic attention shift is synchronized to the saccade, not to fixation point offset. The time-course of attention shift, plotted separately for trials with saccade latencies shorter than the median (across all attentional conditions for that recording session) and saccade latencies longer than the median, is similar when aligned to saccade offset (**a**, **c**) but delayed (**b**, **d**) when plotted aligned to fixation point offset. Time-course for shorter-latency trials in red and longer-latency trials in blue. The attention shift time-course is defined as in Fig. 2a, b as the difference between the firing-rates in the attend-in and attend-out conditions (see Supplementary Fig. 2 for the corresponding PSTHs). Plots show mean difference and s.e.m. Results for Experiment 1 are shown in the top row (**a**, **b**) and for Experiment 2 in the bottom row (**c**, **d**). Red and blue vertical lines in each panel represent the mean time of fixation-point offset (**a**, **c**) or saccade latency (**b**, **d**) for trials with short-latency and long-latency saccades, respectively. Data from both monkeys were pooled for this analysis, and trials were dropped as necessary from each condition so as to match the timing distributions of saccade offset for the attend-in and attend-out conditions for each neuron (also see Text and Supplementary Figs. 5 and 6)

between the attentional modulation of neural activity and behavior in our task will require measuring the neural dynamics of trans-saccadic attention shifts using very brief motion changes.

This rapid recovery of spatial attention following a saccade is consistent with data from double-step saccades[24] and visual search[25] showing that successive saccades can be made with very short inter-saccadic intervals and therefore the target of the second saccade can be rapidly located after the first saccade. In our data, attentional enhancement of the post-saccadic target representation emerged at saccade offset for one monkey and from 50 ms of saccade offset for the other monkey, compared to the approximately 80 ms time reported in a previous study of multi-unit activity in V1 in monkeys performing saccades within a mental curve-tracing task[4]. Khayat et al. calculated the time (of approximately 80 ms) relative to the moment when the stimulus entered the post-saccadic RF. However, given the small size of the RFs in their study (median RF width was 0.94°), this value is likely to be only a few milliseconds longer than the value relative to saccade offset (when the stimulus is at the center of the post-saccadic RF). It therefore appears that the time of emergence of attentional modulation in the post-saccadic target representation in MT in our task is likely earlier, or at least comparable to that reported for the multi-unit data in V1. Also, in our data, attentional enhancement of the pre-saccadic target representation lingered after the saccade and disappeared by 50 ms after saccade offset in one monkey and by 100 ms in the other. This is consistent with previous human psychophysical and imaging evidence suggesting that a lingering attentional modulation of the pre-saccadic target representation for about 100 ms after saccade offset[5–7]. The lingering attentional modulation at the pre-saccadic neuronal representation even after attentional effects have emerged at the post-saccadic neuronal representation is reminiscent of similar effects observed in attentional switch experiments where monkeys covertly changed their locus of spatial

attention while maintaining fixation: in that scenario, attentional effects emerge at the new locus of spatial attention (and the neuronal representation encoding this locus) before attentional effects decay in the neuronal representation encoding the preceding locus of attention[15,26]. The time taken to accomplish the attentional switch in these previous studies in V1, MT and LIP, which can be as short as 150–200 ms[15,26–28], is also comparable to the time-course of the trans-saccadic attention shift in our task. Our analyses, however, showed that trans-saccadic attention shifts are synchronized to saccade planning/execution and are not the direct result of a visual cue-induced attentional switch as in these previous studies. It is possible that covert volitional attentional switches during fixation may engage the same circuitry as attention shifts across saccades. The close co-ordination between peri-saccadic attention shifts and saccadic planning/execution may be facilitated by the overlapping neural circuitry mediating these two phenomena[29,30].

Though both monkeys showed cross-over times close to saccade offset, the data from monkey H show a slightly faster cross-over time than that of monkey E. The attentional cross-over time for monkey H remains faster when calculated (in a similar manner) relative to saccade onset (monkey H: 57 ms, monkey E: 78 ms), indicating that this difference was not an artifact of attention shift dynamics that were linked to saccade onset, with the cross-over time appearing different relative to saccade offset because monkey H had a much shorter saccade duration. This explanation is in any case unlikely because there was only a 1.3 ms difference in mean saccade durations between the two monkeys (monkey H: 28.5 ms, monkey E: 27.2 ms). The difference in attentional cross-over time between the monkeys is also not a result of a delayed sensory response in monkey E (because of a longer sensory latency or differences in RF extent): both monkeys showed the same mean sensory latency to the onset of the RDP (42.5 ms, comparing the response to RDP onset to the

response in the simple-saccade control task where no RDPs appeared. Also, the population average PSTHs (Fig. 2c–f) show a very similar timing for the two monkeys for each individual PSTH, but a relatively early emergence of an attentional effect in monkey H (the difference between the blue and red curves in Fig. 2e emerges earlier than in Fig. 2f). The slightly faster attentional cross-over time in monkey H is therefore not explained by these factors, and appears to be a genuine individual difference between the two monkeys.

We did not find any evidence for attentional enhancement of the post-saccadic target representation before the saccade, even though our experimental design ensured that there was a (distractor) stimulus in the RF before the saccade, and therefore a distractor-driven response before the saccade on which an attentional effect could be seen, if present. This confirms results in two previous studies[18,21] without a stimulus in the RF before the saccade, where no significant attentional modulation of spontaneous activity was found in MT neurons before the saccade. Similarly, in our recent human psychophysical study using a stimulus paradigm very similar to the one here, we did not find any evidence for a predictive, pre-saccadic shift of attention to the post-saccadic target representation[19]. In contrast, a large body of single-neuron recording data from putative attentional control regions in monkeys shows that neurons in the lateral intraparietal area, superior colliculus and frontal eye field[3,10–12,31] as well as some ventral stream areas[32] respond predictively (and sometimes before the saccade) when their RF was stimulated before the saccade, but not after the saccade. Though this anticipatory activity has not been studied explicitly in conditions involving spatial attention, predictive activity is greater for stimuli with greater bottom-up saliency[13] and for stimuli that are learnt visual search targets[14] or saccade targets[13]. Human psychophysical data consistent with an early, pre-saccadic emergence of attentional modulation in the post-saccadic target representation have also been reported[8,9]. We hypothesize that the anticipatory remapping seen in attentional and oculomotor control areas like LIP, frontal eye field (FEF), and superior colliculus (SC) is part of a trans-saccadic attention shift system that maintains optimally timed, saccade-synchronized attention shifts in lower sensory areas like MT. In other words, even though this process starts before the saccade in these areas, its effects in MT, with which these areas are strongly connected[33–35], only manifest after the saccade. In this view, the previous results on trans-saccadic remapping represent the predictive, pre-saccadic shift of attentional pointers on a retinotopic map that keeps track of attended locations across saccades[2], so that attended locations can be preferentially processed with minimal delay after the saccade[18]. This reduction of delay would be especially helpful when planning rapid sequential saccades and could also help maintain an uninterrupted visual experience of attended stimuli across saccades. Additional evidence from other visual areas (for example, in the ventral stream) and using other visual stimuli are undoubtedly needed to resolve these issues. However, the physiological data here, combined with our recent human psychophysics results[19], support our hypothesis that spatial attention and saccadic processing are precisely coordinated to ensure that relevant locations are attentionally enhanced soon after the beginning of each eye fixation and can thus be tracked and rapidly processed across saccades.

## Methods

### Statement on animal research.
Research with non-human primates represents a small but indispensable component of neuroscience research. The scientists in this study are aware and are committed to the great responsibility they have in ensuring the best possible science with the least possible harm to the animals[36]. All animal procedures of this study have been approved by the responsible regional government office (Niedersaechsisches Landesamt fuer Verbraucherschutz und Lebensmittelsicherheit (LAVES)) under the permit numbers 3392 42502-04-13/1100 and 33.14.42502-04-064/07. The animals were group-housed with other macaque monkeys in facilities of the German Primate Center in Goettingen, Germany in accordance with all applicable German and European regulations. The facility provides the animals with an enriched environment (including a multitude of toys and wooden structures[37]), natural as well as artificial light, exceeding the size requirements of the European regulations, and access to outdoor space. Surgeries were performed aseptically under gas anesthesia using standard techniques, including appropriate peri-surgical analgesia and monitoring to minimize potential suffering. The German Primate Center has several staff veterinarians who regularly monitor and examine the animals and consult on procedures. During the study, the animals had unrestricted access to food and fluid, except on the days where data were collected or the animal was trained on the behavioral paradigm. On these days, the animals were allowed unlimited access to fluid through their performance in the behavioral paradigm. Here the animals received fluid rewards for every correctly performed trial. Throughout the study, the animals' psychological and veterinary welfare was monitored by the veterinarians, the animal facility staff and the lab's scientists, all specialized in working with non-human primates. The two animals were healthy at the conclusion of our study and were subsequently used in other studies.

### General methods.
Our description of the Methods here is similar to that presented in our previous publication[18], since the general experimental procedures are the same. We trained two male rhesus monkeys (*Macaca mulatta*, 7–11 kg), monkey H and monkey E, to perform a demanding visuospatial-attention task along with a saccade. Each monkey was implanted with a titanium head holder to minimize head movements during the experiment. One recording chamber was also implanted in each monkey above the left (monkey E) or the right (monkey H) parietal cortex to allow access to MT, with implantation locations chosen based on a preceding MRI scan. Monkey H and monkey E were around 15 and 9 years old, respectively, during the period of recording.

The experiments were performed in a dimly lit room with the only source of light being the display monitor. A CRT monitor (Sony Trinitron GDM-FW900) at a distance of 57 cm from the monkey was used to display the visual stimulus at a refresh rate of 76 Hz and a spatial resolution of 40 pixels degree$^{-1}$. The monkey sat in a custom-made primate chair during the experiment. Stimulus presentation, reward delivery, electrophysiological and behavioral data collection was controlled by custom software and run on an Apple Macintosh computer. All stimulus onsets and durations were specified in terms of number of frames (CRT monitor refreshes), and the stimulus presentation times reported here in millisecond units are correct to within 13 ms (the duration of one frame), given the vertical scan-rate properties of the CRT monitor. The animals received a fluid reward immediately following each correct trial. The eye-position was monitored by an EyeLink 1000 (SR Research, Canada) system at 1000 Hz. Neuronal activity was recorded extracellularly with a 5-channel micro drive system (Mini Matrix, Thomas Recording, Giessen, Germany) and processed using the Plexon data acquisition system (Plexon Inc., Dallas, TX). Only data from well-isolated neurons were used for the analysis. MT was identified by referencing the recordings to the structural MRI and by the physiological properties of the recorded neurons: most neurons were direction-tuned, the average diameter of the RFs was approximately equal to the RF eccentricity and there was a predictable progression of RF centers at different locations along the superior temporal sulcus.

### Behavioral tasks and stimuli.
Once a neuron was isolated, we had the monkey perform a fixation task where the monkey had to maintain fixation on a fixation point and respond to a brief luminance change at the fixation point. During this fixation period, we located the RF by moving a stationary circular RDP across the screen using a mouse. We then determined the neuron's preferred direction and speed during the fixation task by presenting a RDP with moving dots within a circular aperture in the RF, changing the direction and speed every 250 ms to a value picked from a set of 3 possible speeds (4, 8, or 16 degrees s$^{-1}$) and 12 possible directions (evenly separated by 30° around a circle). For the main experiment, we used stimuli with directions and speeds equal or close to the preferred direction and the preferred speed thus determined.

After identifying the RF location and preferred direction, we switched to the main experiment (Fig. 1), where the monkey performed an experimental "attention-saccade" task (either Experiment 1 or Experiment 2) and a control task in a pseudo-randomly interleaved manner. In the experimental task, comprising 80% of trials, the monkeys were trained to concurrently perform a visuospatial attention task and a saccade task on each trial: they were instructed to pay attention to the target RDP and make a saccade if the fixation point jumped to a new location. The monkeys initiated the trial by holding a metal bar and foveating a black fixation point. After 118 ms of fixation, to indicate the location of the upcoming target RDP, a stationary circular RDP cue (of the same size as the target) was presented for 263 ms. After an additional delay of 329 ms following cue offset, four moving RDPs (2° in radius, all dots moving at or close to the neuron's preferred direction of motion and within stationary circular apertures) were presented on the screen. Two of the RDPs were presented in the neuron's pre-saccadic and post-saccadic RFs respectively and the other two RDPs were located opposite to these stimuli (i.e., reflected across the horizontal or vertical meridian, see Fig. 1). The monkeys' task was to respond to a transient (132 ms) direction

change in the RDP at the previously cued location (the target) by releasing the metal bar (within 600 ms of the change), but ignore similar changes in the distractor (the RDP opposite to the target). Trials terminated 600 ms after the target change, with the monkey receiving a drop of juice if the bar had been correctly released during this period. In addition, during the trial, if the fixation point jumped to a new location, the monkeys had to re-fixate the fixation point while continuing to attend to the cued target. The direction change in the target RDP could occur between 263 ms to 1973 ms after RDP onset; the timing led to a near-constant hazard function for target-change around the time of the saccade (Supplementary Fig. 9). Distractor changes occurred on about 37% of trials and never more than once on each trial. The timing of distractor changes overlapped that of target changes, with the additional requirement that any distractor change occurred at least 500 ms before the target change on each trial. This separation ensured that the monkeys' rare responses to the distractor change could be easily identified and distinguished from their responses to the target change. The fixation point jumped to its new location (and became the saccade target) 1382 ms after fixation point onset (i.e., 671 ms after RDP onset); however, this event did not occur if the trial had terminated by then (either by a correct or incorrect bar-release or by a missed target change). The saccade target then stayed on for 1368 ms (or until the end of the trial). There was a one-frame (13 ms) overlap between the fixation point and the saccade target, so that the fixation point disappeared one frame after the saccade target appeared: perceptually, the fixation point appeared to jump from its original location to the saccade target. Once the fixation point jumped, the monkey had to make a saccade to the new location of the fixation point within 263 ms and maintain fixation until the end of the trial. The saccade target appeared between 10° and 20° eccentrically (value fixed for each neuron, and either 15° or 20° in most cases). Saccades were always either horizontal or vertical. We used a fixed and predictable time for the fixation point jump to reduce the temporal uncertainty about when the fixation point would jump and thereby minimize the monkeys' need to monitor the fixation point or saccade target location in order to detect the saccade jump. This would enable the monkeys to better focus their attention on the target RDP. The median saccade latency was 136 ms in monkey H and 142 ms in monkey E; 99% of the saccades occurred before 217 ms in monkey H and 229 ms in monkey E.

In Experiment 1, the cue (and by extension, the target RDP) was located either in the neurons' pre-saccadic RF (attend-in condition) or opposite to it (attend-out condition) equally often in a pseudo-randomly interleaved manner. Experiment 2 was similar, except that the cue (and by extension, the target RDP) was located either in the neurons' post-saccadic RF (attend-in condition) or opposite to it (attend-out condition) equally often in a pseudo-randomly interleaved manner. The control task, comprising 20% of trials, was a 'simple-saccade' task where the monkey simply made a saccade when the fixation point jumped to a new location and maintained fixation till the end of the trial to obtain the juice reward. There was no concurrent attentional task; i.e., no cue and no moving RDPs were presented. Data from this control task were only used to select visually responsive neurons for further analysis (see below).

In all the tasks, the background was always gray with a luminance of 14.2 cd m$^{-2}$, and the fixation point and RDPs including the stationary cue were black with the luminance of 0.68 cd m$^{-2}$. Individual RDP dot size was 0.1° × 0.1°, and the dot density was 10 dots deg$^{-2}$. The fixation point and the saccade target were both squares (length of each side = 0.3°). Monkeys had to maintain fixation within a circular window of 2° radius around the fixation point before the fixation point jumped. Following a period of 263 ms after the fixation point jumped (which gave the monkeys time to make the saccade), the monkeys had to maintain fixation within a circular window of 3° radius around the saccade target. For the analysis, we used a window that started from 20 ms after saccade offset (see Data Analysis section below)[38,39]. The saccade direction was set according to the position of the RF: for example, if the RF center is directly above or below the fixation point, we used a horizontal saccade, while if the RF center is directly to the left or right of the fixation point, we used a vertical saccade. If the RF center was offset both vertically and horizontally from the fixation point, the choice was no longer critical, but we usually used a horizontal saccade.

**Data analysis**. All data analysis was performed using custom software in MATLAB (MATLAB Inc, Natick, MA). We detected saccades using a velocity-threshold criterion that was validated by visual inspection. Onset (and offset) times were determined by when the eye velocity exceeded (and then dropped below) 100° s$^{-1}$. This threshold value was set to lie clearly above the peak excursions of the baseline noise in the eye-velocity traces, and the algorithm was validated by visual inspection for each monkey. By considering the saccade to have ended when the velocity dropped below a threshold value well above the baseline noise (and when the eye was still moving), our threshold criterion provides a conservative, i.e., early definition of saccadic end-point and therefore if anything, a longer estimate of the cross-over time for the trans-saccadic attention shift. Except when explicitly otherwise specified, throughout this paper, we analyze data aligned to saccade offset, and use the term "saccade latency" to refer to the time interval from the disappearance of the fixation point (the saccade cue) to the end of the saccade.

We included data from all neurons that showed a visual response to the RDP in the RF both before and after the saccade. We identified these neurons as those that showed a significantly greater postsaccadic response (one-sided $t$-test, $p < 0.05$, Bonferroni-corrected) in the attention-saccade task compared to the simple-

saccade control in the time-periods 0 to 600 ms following RDP onset (i.e., they were visually responsive to the RDP in the pre-saccadic RF) and 0 to 600 ms following the saccade (i.e., they were visually responsive to the RDP in the post-saccadic RF). In total, we analyzed 84 neurons in Experiment 1 (56 in monkey H and 28 in monkey E) and 84 neurons in Experiment 2 (52 neurons in monkey H and 32 neurons in monkey E). Their RF eccentricities ranged from 6.7° to 14.9° in Experiment 1 and 5.1° to 16.4° in Experiment 2. Twenty-nine neurons (all in monkey H) provided data for both Experiments 1 and 2. There were at least nine trials for both the attend-in and attend-out conditions in Experiment 1 and at least 14 trials in Experiment 2 in all neurons. After excluding fixation breaks, monkey H performed the task correctly on 93.2% of trials (with "false-positive" releases on 3.7% and "misses" on 3.1% of trials), while monkey E performed the task correctly on 84.6% of trials (with false-positive releases on 8.8% and misses on 6.6% of trials). We analyzed all correctly completed trials where the target change did not occur too close to saccade offset (i.e., the target change was at least 200 ms earlier or 200 ms later than saccade offset), since this was the time-period that we focused on to measure the time-course of the trans-saccadic attention shift. Additionally, trials were only included if the eye stayed within 3° of the final eye-fixation position from 20 ms after the saccade offset to the end of the trial; the 20 ms period served to take post-saccadic dynamic overshoots and glissades into account[38,39]. We calculated the final eye-fixation position by taking the median eye-position from 125 ms after the saccade end to the target change time across trials as the final eye-fixation position (this corrected for small eye-calibration errors). Using a more stringent criterion where the eye had to stay within 2° of the final fixation position, and the saccade had to end within 2° of the final fixation position, and where no additional (small) saccades were made during the critical analysis period (0 to 100 ms following saccade offset) also gave a very similar time-course for the attention shift and did not affect our conclusions (Supplementary Fig. 4). Given the complex nature of our main task, as well as the difficulties of mapping the edges of the RF, we did not always map the RF size before starting the main task: we only ensured that the post-saccadic stimulus in Experiment 1 lay outside the pre-saccadic RF (and vice-versa for Experiment 2), even with the slight variability in saccadic endpoints. We also ensured that the RF locations shifted commensurately with fixation position (i.e., the RFs were retinotopic).

Peri-stimulus time histograms (PSTHs) were calculated by filtering the spike-trains (recorded with 1 ms resolution) with a truncated Gaussian window (standard deviation 15 ms; filter width 100 ms) stepping every 1 ms: the filter was centered at each time-step. We have also analyzed the data with standard-deviation values of 10 ms, and our conclusions remain unchanged. The mean activity for each neuron across trials was first calculated and then these mean PSTHs for individual neurons were averaged across neurons to obtain the displayed PSTHs. When discussing the results of bin-by-bin statistical significance tests (Fig. 2a, b), if a test is significant/non-significant for a given bin, we assume that it is significant/non-significant for the entire duration of the bin. In the attend-out conditions, to avoid the transient response to the brief change in the distractor stimulus within the RF, we excluded the period of 0 to 300 ms following the distractor change from the PSTH and firing-rate calculations. To calculate percentage changes in firing-rate when comparing the two conditions in Fig. 1, we followed the conventional method and first calculated a modulation index for each neuron as the difference in the firing-rates for the two conditions divided by their sum, averaged the modulation indices and then converted the average into a percentage change. We used two-sided $t$-tests throughout unless mentioned otherwise; using the signed-rank test produced similar results. For the analysis in Fig. 3, for each session, trials were dropped from one or both attentional conditions in order to match the distribution of saccade latencies (within 20 ms bins): overall, 66% of trials were retained after matching. To get an estimate of the variability of the attentional cross-over time, we used a bootstrap procedure where, for each of 15,000 bootstrap repetitions, we randomly sampled $N$ neurons with replacement from the original dataset (which had $N$ neurons), and from each sampled neuron (with $n$ trials), we then randomly sampled $n$ trials with replacement to generate a new simulated dataset. We then calculated the cross-over time for this simulated dataset. This process was repeated 15,000 times to create a bootstrap distribution of cross-over times, and the IQR was used to summarize the variability of the bootstrap distribution.

**Data availability**. The data and analysis code that support the findings of this study are available from the corresponding author upon reasonable request.

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

## Acknowledgements

The study was funded by grants to S.T. from the German-Israeli Foundation for Scientific Research and Development (GIF): 1108-79.1/2010, the Federal Ministry of Education and Research (BMBF) Germany: 01GQ1005C and from the German Research Foundation (DFG): Research Unit 1847 "Physiology of Distributed Computing Underlying Higher Brain Functions in Non-Human Primates" (Project A01).

## Author contributions

T.Y. and B.S.K. designed the study; T.Y. conducted the experiments; T.Y. and B.S.K. analyzed the data; T.Y., S.T., and B.S.K. wrote the manuscript.

## Additional information

**Competing interests:** The authors declare no competing interests.

