## [Peer Review File · Nature Communications]

Reviewers' comments:

Reviewer #1 (Remarks to the Author):

This is a paper about the timing of attentional shifts in the visual cortex during saccades. The authors make use of a paradigm that controls the locus of attention in spatial coordinates while monkeys make saccades that shift the positions of stimuli on the retina. Recordings from single neurons in MT show that attentional shifts are synchronized with the execution of the saccade, so that they appear at the post-saccadic retinal position within 10s of milliseconds after the saccade. The results thus show that attention tracks the retinal positions of attended objects with high temporal fidelity.

The work builds on previous findings from the same lab, which documented the basic phenomenon of attentional shifts during saccades. A modification of the previous paradigm has allowed the authors to measure the precise timing of the attention shift from the presaccadic to the postsaccadic retinal location. While it is obvious that the shift had to occur, the timing could conceivably have been very different, either starting before the saccade, as suggested by the phenomenon of predictive remapping, or taking some time to materialize after the saccade, as suggested by the general sluggishness of attention in other contexts. Thus I think that the finding of tight synchronization is a potentially valuable contribution to the larger literature on saccades, vision, attention, etc. Another strength of the paper is that the paradigm is elegantly designed to separate attentional shifts from other influences due to the saccade and retinal inputs. Consequently the analysis ends up being quite straightforward, and the presentation is easy to understand.

Comments for revisions:

1. The main result is that the attentional modulation for the post-saccadic stimulus overtakes that for the pre-saccadic stimulus shortly after the saccade. I find this to be generally convincing, but there are a few sources of uncertainty that should be clarified or explored. First, if I understand the methods correctly, Experiments 1 and 2 were performed in separate blocks and possibly on different neurons? Second, given the sequence of events in each trial, the hazard functions must be rather different for the two experiments, which should lead to different types of attentional modulation (e.g, Ghose & Maunsell). Both considerations suggest that a comparison of the simple firing rate indices used in Figure 2 is not appropriate, since it would include offsets due to potentially different firing rates and behavioral states. I don't have any reason to think that this will affect the results categorically, but it should be easy to check.

2. While the paper does document that the attentional shift is synchronized to the saccade, the functional implications of this finding are a bit unclear. Early in the postsaccadic period the modulation is on the order of a few spikes per second, which for the early postsaccadic period, would translate into zero or one spikes for most neurons on most trials. In both monkeys, the modulation appears to grow through time (possibly tracking the hazard function). Can the authors say anything about the animals' behavior through time? Does performance on the attention task relate to the neural modulation in any way? The problem is that it's not clear how useful the attentional shift can be if it takes hundreds of milliseconds to develop after the saccade.

Cosmetic point:

3. I found it difficult to work out the timing of the different events in the trial, particularly the attentional cue (RDP change) and reaction (lever release) relative to the saccade timing, which is crucial. Perhaps the authors can add this information schematically to Figure 1.

Reviewer #2 (Remarks to the Author):

Review of

Saccade-synchronized rapid attention shifts in macaque visual cortical area MT

Yao, Treue & Krishna

This paper describes a straightforward study with two simple and tightly linked experiments. It is clearly written and presents a convincing result: Attentional modulation of the activity of MT neurons crosses over from a presaccadic to a post-saccadic spatial representation shortly after the saccade is executed. This means that, unlike more frontal neurons, MT neurons do not exhibit any predictive presaccadic modulations, but instead appear to closely reflect where the relevant stimulus is. I strongly support publication of this interesting and clearly described finding.

I do have a few comments and suggestions:

1) I miss data on the precise landing position and how well the RFs were centered on the stimulus after the saccade. Usually the first saccade to a target undershoots and monkeys make one or more corrective saccades before gaze is centered on a new fixation point. Did these small successive RF shifts influence the measurement of the timing of the postsaccadic attentional modulation in experiment 2?

- What was the size of the RFs relative of the size of the moving patches and what was the moment that they were aligned with the stimulus? Did they always immediately fall on the stimulus after the saccade? Was the overlap between RF and stimulus always 100%?

- Can the authors illustrate the size of the typical RF and stimulus?

2) The monkeys had to monitor the moving dots for a change in motion direction. What was the precise timing distribution of these changes? Uniform? Exponential? Although the authors state that the task was not designed to look at the dynamics of attention during the saccade, some information about how the accuracy was before and after the saccade would strengthen the paper.

- In the previous paper by Yao et al (eLife, 2016) there is a noticeable drop in accuracy at the locus of the target just before the eye movement, which was thought to be caused by diverting attention to the jumping fixation point. However, in Fig. 1c and Fig. 2 the modulation remains relatively uninfluenced by the saccade planning. Please discuss this apparent discrepancy.

3) I don't understand why the authors analyse their data at such a coarse temporal resolution when they are interested in the temporal aspects of the attentional modulation. PSTHs and difference plots are calculated and displayed based on 25 or 50 ms non-overlapping bins, yet effect latencies and crossover moments are given with ms resolution. Sometimes they resort to interpolating data traces between these bin centers. Why not use sliding windows with steps of a few ms? I'm not sure if it would change anything qualitatively, but I don't immediately see the rationale behind their choice.

4) It has been shown that attention not only affects the magnitude of a response but also its variability (e.g., Mitchell, Sundberg & Reynolds, 2007 Neuron). Would it be possible to look at the presaccadic effect of a variability measure such as the Fano factor?

5) p7. The attentional enhancement is compared between the pre- and postsaccadic time-window by performing t-tests on the activity in both windows. Is this also done in a single 2-way ANOVA with factors time window (pre/post) and attentional state (attended/unattended)? The statement "a direct comparison between the attentional effects in the two time periods also showed that that the attentional effect in the presaccadic period was greater than that in the postsaccadic ..." is abit ambiguous. What is tested here and how? A significant interaction effect in the ANOVA as mentioned above is expected, but I'm not sure that's what was done.

6) To explain the difference between the present results and previous studies reporting pre-saccadic RF shifts, the authors suggest that there are two separate systems: one that predicts the consequence of an eye movement (including e.g. LIP) and one that reflects that as change in attentional modulation, locked to the saccade (including MT). It seems that an alternative explanation might be that the studies without pre-saccadic shifts, there was a distractor item in the RF that needed to be suppressed, whereas in the studies with pre-saccadic RF shifts there typically was no item in the neuron's RF before the saccade. Or are there reasons to reject this as an alternative explanation?

7) There are peaks visible in Fig. 3b, which are not visible in Fig. 3a. Fig. S2 seems to reveal that there are timing differences between saccades in the two attention conditions. The authors may want to point out the cause of the two peaks in Fig. 3b. Furthermore, were there indeed systematic difference in saccadic latency between the attention conditions? What caused them? Can this have influenced the conclusions?

Minor

1) p12. Ln14-16. I understand the message, but I get lost in this sentence. There may be a word missing (around 'sufficient'), but I think this could be rewritten more clearly.

2) I think there's something going wrong with the panel referencing in the legend of Sup fig 2. The legend talks about short and long latency saccades in panel a-d and e-h respectively, but I guess this should be a-b,e-f and c-d,g-h.

3) On p. 9, line 3 the reader wonders what these IQRs are and only finds out in the methods that they are related to bootstrapping. This information should be added when the IQRs are first mentioned.

4) Legend Fig. 2, line 15: the curves in c and d are magenta and cyan, not red and blue.

5) Is there a relation between the latency of postsaccadic attentional modulation and that at presaccadic stimulus onset? I realize that this is only possible for the neurons that have been recorded in both experiments.

Response to the Reviewers

Reviewer #1 (Remarks to the Author):

This is a paper about the timing of attentional shifts in the visual cortex during saccades. The authors make use of a paradigm that controls the locus of attention in spatial coordinates while monkeys make saccades that shift the positions of stimuli on the retina. Recordings from single neurons in MT show that attentional shifts are synchronized with the execution of the saccade, so that they appear at the post-saccadic retinal position within 10s of milliseconds after the saccade. The results thus show that attention tracks the retinal positions of attended objects with high temporal fidelity.

The work builds on previous findings from the same lab, which documented the basic phenomenon of attentional shifts during saccades. A modification of the previous paradigm has allowed the authors to measure the precise timing of the attention shift from the presaccadic to the postsaccadic retinal location. While it is obvious that the shift had to occur, the timing could conceivably have been very different, either starting before the saccade, as suggested by the phenomenon of predictive remapping, or taking some time to materialize after the saccade, as suggested by the general sluggishness of attention in other contexts. Thus I think that the finding of tight synchronization is a potentially valuable contribution to the larger literature on saccades, vision, attention, etc. Another strength of the paper is that the paradigm is elegantly designed to separate attentional shifts from other influences due to the saccade and retinal inputs. Consequently the analysis ends up being quite straightforward, and the presentation is easy to understand.

We thank the reviewer for the accurate summary and positive appraisal.

Comments for revisions:

1. The main result is that the attentional modulation for the post-saccadic stimulus overtakes that for the pre-saccadic stimulus shortly after the saccade. I find this to be generally convincing, but there are a few sources of uncertainty that should be clarified or explored. First, if I understand the methods correctly, Experiments 1 and 2 were performed in separate blocks and possibly on different neurons? Second, given the sequence of events in each trial, the hazard functions must be rather different for the two experiments, which should lead to different types of attentional modulation (e.g, Ghose & Maunsell). Both considerations suggest that a comparison of the simple firing rate indices used in Figure 2 is not appropriate, since it would include offsets due to potentially different firing rates and behavioral states. I don't have any reason to think that this will affect the results categorically, but it should be easy to check.

First question: Yes, as we mention in the Methods (page 29 in the resubmission), “In total, we analyzed 84 neurons in Experiment 1 (56 in monkey H and 28 in monkey E) and 84 neurons in Experiment 2 (52 neurons in monkey H and 32 neurons in monkey E). 29 neurons (all in monkey H) provided data for both Experiments 1 and 2.”

Second question: Experiments 1 and 2 had identical timing distributions for target and distractor changes, and therefore identical hazard functions. Further, our comparison of the time-course of peri-saccadic attentional decay (in Experiment 1) and peri-saccadic attentional emergence (in Experiment 2) are made in contemporaneous time-windows. Within the 200 ms window around saccade offset, the attentional effect for Experiment 2 starts around zero and rises above zero after the saccade. But, this cannot be ascribed to a hazard function effect, because over this same period of time, the attentional effect for Experiment 1 starts above zero and drops down to zero (and below the curve for Experiment 1) after the saccade. Further, the hazard function changes only minimally over the 200 ms period surrounding the mean time of saccade offset: we now show this with a plot of the instantaneous probability and hazard function for the target-change as Supplementary Figure 7. We thank the Reviewer for bringing this issue up.

The Reviewer states that “a comparison of the simple firing rate indices used in Figure 2 is not appropriate, since it would include offsets due to potentially different firing rates and behavioral states. I don’t have any reason to think that this will affect the results categorically, but it should be easy to check”. We are not entirely sure about which precise analysis the Reviewer considers appropriate, or how the Reviewer would like us to check for appropriateness. However, we guess that the Reviewer is, in general, concerned about how to take the effects of potential differences in baseline firing-rate (between the neurons recorded in Experiments 1 and 2) into account when evaluating the observed attentional effects. We respond to this as follows –

- a) As we discussed in the manuscript, the mean firing-rates of the recorded population for Experiment 1 and 2 were very different for monkey E, but quite similar for monkey H. In the original submission, we therefore performed the analysis in Supplementary Figure 1 (of the original submission) for monkey E where we dropped the highest firing-rate neurons from Experiment 2 to better match the mean firing-rates between the two experiments: the estimated time-course of attentional effects in Experiment 2 remained essentially unchanged as a result, and the median cross-over time changed only minimally. We have appended that original Supplementary Figure 1 to this response. In the new submission, we have replaced this analysis with a similar analysis that applies to both monkeys. Instead of truncating the upper 25 % of firing rates, we have now “mean-matched” the firing-rate distributions for Experiments 1 and 2 in both monkeys (using 5 Hz bins) and show that our conclusions remain unchanged as a result (Supplementary Figure 1).
- b) In Supplementary Figure 2, we now also show the results using a ratio measure rather than a difference measure and obtain a very similar estimate for the cross-over time and similar dynamics of attention shifts. The motivation for this analysis is as follows: if the

effect of attention is multiplicative in the immediate post-saccadic period, such a measure would estimate the multiplicative factor. It is well-known from the ratio estimator literature in statistics that a reasonable first-pass estimator for the ratio of the population means of two populations A and B (μ_A/μ_B) is given by $\text{mean}(a)/\text{mean}(b)$ where a and b are the samples from A and B (http://en.wikipedia.org/wiki/Ratio_estimator and references therein). Though this estimator can be easily shown to slightly under-estimate the true ratio, it performs better than other candidate estimators (including the naive $\text{mean}(a/b)$ estimator that averages the ratios for each sample). The numerous corrections that have been proposed do not necessarily improve the situation in all circumstances (because they only sometimes correct the bias or only do so asymptotically, do not perform as well for small samples and often have much higher variance). In our own simulations (using a random multiplicative effect on a population of Poisson neurons), the $\text{mean}(a)/\text{mean}(b)$ estimator substantially outperforms the conventional attentional index (which involves calculating $r = \text{mean}((a-b)/(a+b))$ and then back-converting to a ratio as $(1+r)/(1-r)$) and also outperforms or has similar performance to all the corrected estimators we tested (since the bias is very small).

2. While the paper does document that the attentional shift is synchronized to the saccade, the functional implications of this finding are a bit unclear. Early in the postsaccadic period the modulation is on the order of a few spikes per second, which for the early postsaccadic period, would translate into zero or one spikes for most neurons on most trials. In both monkeys, the modulation appears to grow through time (possibly tracking the hazard function). Can the authors say anything about the animals' behavior through time? Does performance on the attention task relate to the neural modulation in any way? The problem is that it's not clear how useful the attentional shift can be if it takes hundreds of milliseconds to develop after the saccade.

As we discussed in our original submission (and in the new one: page 13, upper paragraph), a limitation of the current manuscript is that brief target-changes were not used (in contrast to our eLife paper with human subjects). So we agree with the Reviewers' general point that the manuscript does not correlate the observed attentional modulation with the behavioral performance. Using brief target changes is essential to analyze the functional implications of the observed changes and is an important target for future work. Due to the relatively long-duration target-changes we used, performance was not significantly modulated around the saccade: in fact, we do not even observe the expected effect of saccadic suppression (Supplementary Figure 8).

We would however argue against the Reviewer's second contention that a modulation of "the order of a few spikes per second" may not have functional consequences over the "early post-saccadic period". We think that the observed attentional effects (within a 50 ms window) are very plausibly sufficient to produce behavioral effects, in the sense that the attended location can be perfectly decoded on every trial with a realistic number of neurons. We show this in a figure

appended to this response (Reviewer Response Figure 1). We show that with the observed attentional effect sizes, which indeed translate to a mean modulation of less than 1 spike within a 50 ms window, less than 1000 independent Poisson neurons are sufficient to near-perfectly decode the attended location. We would argue that given a very rough estimate of 100 neurons per mini-column in rhesus macaque cortex, and the 4 degree diameter of the target stimulus, as well as the relatively low degrees of correlation in cortex, it is plausible that 1000 MT neurons (behaving roughly as independent Poisson neurons) are available to the brain on this task. Small, but reliable effects can of course be magnified downstream to produce large effects, as most notably and directly shown by the effects of cortical single-neuron stimulation (Doron and Brecht 2015). If the Reviewers/Editor so wish, we can include this argument and figure in the manuscript as Supplementary Information.

Finally, as we discuss above, the neural modulation is unlikely to be related to the hazard function.

Details of the simulation presented in Reviewer Response Figure 1: We started with 4 Poisson populations: the number of neurons in each population was varied systematically across simulation conditions from 10 to 1000 (X-axis in the figure). In each simulation, we simulated 10,000 trials. On each trial, the baseline mean firing-rates were chosen randomly for each neuron in the four Poisson populations from a uniform distribution (minimum: 30 spikes per second = 0.15 spikes in 50 ms, maximum: 100 spikes per second = 5 spikes in 50 ms). The firing-rate for the target RDP representation was then enhanced by an attentional effect. This attentional effect was estimated for the two monkey separately, and in two ways: as a difference (Figure 2 a-b) and as a ratio (Supplementary Figure 2 a-b), giving the four different plots in the figure ((solid lines: monkey H and dashed lines: monkey E; difference with circular markers and ratio with asterisk markers). When the attentional effect was estimated as a difference in firing-rates, the effect was added to the baseline firing-rate of the target RDP population, while when the attentional effect was estimated as a ratio, the baseline firing-rate of the target RDP population was multiplied by the estimated effect. The number of spikes was drawn randomly using these mean firing-rates from a Poisson distribution, and the population that produced the largest number of spikes was declared the winner on that trial. (To estimate the attentional effects from Figure 2 and Supplementary Figure 2 that we used in the simulation, we subtracted the attentional effect for Experiment 1 from the attentional effect for Experiment 2 in the 50-100 ms bin following saccade offset. The actual effects used in the simulation were – a difference of 9.66 spikes per second and a ratio of 1.26 (26 percent enhancement) for monkey H, and a difference of 4.18 spikes/second and ratio of 1.0726 (7.26 percent enhancement) for monkey E. We emphasize that these choices are very rough and are by no means critical to our conclusion. Assuming independent Poisson neurons, the conclusion that very accurate attentional decoding can be achieved on each trial using our observed effect sizes and less than 1000 neurons is very robust to variations in how we choose to calculate and implement these effect sizes.

Cosmetic point:

3. I found it difficult to work out the timing of the different events in the trial, particularly the attentional cue (RDP change) and reaction (lever release) relative to the saccade timing, which is crucial. Perhaps the authors can add this information schematically to Figure 1.

We thank the Reviewer for this suggestion and have added this information to Figure 1. We hope the Reviewer finds this satisfactory.

Reviewer #2 (Remarks to the Author):

*Review of Saccade-synchronized rapid attention shifts in macaque visual cortical area MT
Yao, Treue & Krishna*

This paper describes a straightforward study with two simple and tightly linked experiments. It is clearly written and presents a convincing result: Attentional modulation of the activity of MT neurons crosses over from a presaccadic to a post-saccadic spatial representation shortly after the saccade is executed. This means that, unlike more frontal neurons, MT neurons do not exhibit any predictive presaccadic modulations, but instead appear to closely reflect where the relevant stimulus is. I strongly support publication of this interesting and clearly described finding.

We thank the Reviewer for the accurate summary and positive appraisal.

I do have a few comments and suggestions:

1) I miss data on the precise landing position and how well the RFs were centered on the stimulus after the saccade. Usually the first saccade to a target undershoots and monkeys make one or more corrective saccades before gaze is centered on a new fixation point. Did these small successive RF shifts influence the measurement of the timing of the postsaccadic attentional modulation in experiment 2?

- What was the size of the RFs relative of the size of the moving patches and what was the moment that they were aligned with the stimulus? Did they always immediately fall on the stimulus after the saccade? Was the overlap between RF and stimulus always 100%?

- Can the authors illustrate the size of the typical RF and stimulus?

We have now added an analysis of these issues to the manuscript. As we state in the Supplementary Information (page 2): “We first calculated a final eye-fixation position by taking the median eye-position from 125 ms after the saccade end to the target change time across trials as the final eye-fixation position (this corrected for small eye-calibration errors). 98.9 and 98.2 % of saccades ended within 3 degrees of this final eye-position in monkeys H and E respectively; the values for a 2 degree range were 97.1 % and 89.7 %, and for a one-degree range were 72.6 % and 52.6 %. The saccades in monkey H thus ended closer to the final fixation position than in monkey E. Both monkeys showed additional changes in eye-position (drifts and small saccades) after acquiring the saccade target: this is not surprising given that they were performing a dual task (fixate and attend to the peripheral target RDP to detect a small change). Consistent with the better saccade accuracy in monkey H compared to monkey E, the first corrective saccade (of larger than 1 degree amplitude to avoid small microsaccades and dynamic overshoots) in monkey H ended only 0.04 degrees closer on average to the final fixation position (SEM=0.016 degrees) compared to its starting position. In monkey E, this value was 0.8 degrees (SEM=0.046 degrees). A similar result was obtained when calculating over all saccades during the post-saccadic fixation period: these saccades ended 0.19 degrees (SEM=0.01 degrees) closer in monkey H and 0.81 degrees (SEM=0.04 degrees) closer in monkey E. We also note that the accuracy was also quite consistent across attention conditions and thus accuracy differences do not impact our results: the mean difference between the saccade accuracy in the attend-in and attend-out conditions was very small (Experiment 1, monkey H: 0.07 degrees (SEM=0.04 degrees), monkey E: -0.005 degrees (SEM=0.03 degrees); Experiment 2, monkey H: 0.07 degrees (SEM=0.03 degrees), monkey E: -0.13 degrees (SEM=0.05 degrees)).”

We have now edited the manuscript and adjusted our analyses to take these concerns into account. As we now state in the Methods (page 30): “Additionally, trials were only included if the eye stayed within 3 degrees of the final eye-fixation position from 20 ms after the saccade offset to the end of the trial; the 20 ms period served to take post-saccadic dynamic overshoots and glissades into account (Bahill, Clark et al. 1975, Nystrom, Hooge et al. 2013). We calculated the final eye-fixation position by taking the median eye-position from 125 ms after the saccade end to the target change time across trials as the final eye-fixation position (this corrected for small eye-calibration errors). Using a more stringent criterion where the eye had to stay within 2 degrees of the final fixation position, and the saccade had to end within two degrees of the final fixation position, and where no additional (small) saccades were made during the critical analysis period (0 to 100 ms following saccade offset) also gave a very similar time-course for the attention shift and did not affect our conclusions (Supplementary Figure 4).”

Finally, the Reviewer asked about the size of the RFs relative to the size of the RDPs. As we now state in the Methods section (page 29, lines 1-2) – “The RF eccentricities ranged from 6.7 to 14.9 degrees in Experiment 1 and 5.1 to 16.4 degrees in Experiment 2.” Given the complex nature of our main task, as well as the difficulties of mapping the edges of the RF, we did not always map the RF size before starting the main task: we only ensured that the post-saccadic

stimulus in Experiment 1 lay outside the pre-saccadic RF (and vice-versa for Experiment 2) and that the RF locations shifted commensurately with fixation position (i.e. the RFs were retinotopic). However, the classical estimates of RF size in MT indicate an average diameter for the RF that is equal to the eccentricity. Thus, since our RDPs were 4 degrees in diameter, the RFs were likely to be usually much larger than the RDPs. However, given that we did not map the RF size explicitly, we prefer not to indicate anything explicit about the size of the “typical RF” relative to the stimulus.

2) The monkeys had to monitor the moving dots for a change in motion direction. What was the precise timing distribution of these changes? Uniform? Exponential? Although the authors state that the task was not designed to look at the dynamics of attention during the saccade, some information about how the accuracy was before and after the saccade would strengthen the paper.

- In the previous paper by Yao et al (eLife, 2016) there is a noticeable drop in accuracy at the locus of the target just before the eye movement, which was thought to be caused by diverting attention to the jumping fixation point. However, in Fig. 1c and Fig. 2 the modulation remains relatively uninfluenced by the saccade planning. Please discuss this apparent discrepancy.

- a) As we mention in our response to Reviewer 1, we now show the timing distribution of the target changes as well as the hazard function in Supplementary Figure 7.
- b) The Reviewer’s observation is spot on. We now include this text in the Discussion (page 13): “We do note that the reduction in detection performance around the saccade cue onset, that was visible in the human behavioral data, is not visible in the MT activity we record here: one possibility is that the observed reduction in performance around saccade cue onset is mediated by areas other than MT. This would be consistent with a recent study showing that inactivating the superior colliculus can lead to large behavioral deficits in a very similar task to ours, without similarly affecting MT and MST activity (Zenon and Krauzlis 2012).”

3) I don’t understand why the authors analyse their data at such a coarse temporal resolution when they are interested in the temporal aspects of the attentional modulation. PSTHs and difference plots are calculated and displayed based on 25 or 50 ms non-overlapping bins, yet effect latencies and crossover moments are given with ms resolution. Sometimes they resort to interpolating data traces between these bin centers. Why not use sliding windows with steps of a few ms? I’m not sure if it would change anything qualitatively, but I don’t immediately see the rationale behind their choice.

We have adopted the Reviewer’s recommendation: we now consistently filter the spike-trains (recorded with 1 ms resolution) with a truncated Gaussian window (standard-deviation 15 ms;

filter width 100 ms) stepping every 1 ms. Using a 10 ms standard-deviation filter led to noisier curves, but did not change the cross-over time substantially (as we now report in the Results).

4) It has been shown that attention not only affects the magnitude of a response but also its variability (e.g., Mitchell, Sundberg & Reynolds, 2007 Neuron). Would it be possible to look at the perisaccadic effect of a variability measure such as the Fano factor?

We appreciate the Reviewer's idea and have performed similar Fano factor analyses ourselves before ((Falkner, Goldberg et al. 2013), but for various reasons, including the complexities of calculating and interpreting Fano factors over smaller time-windows (say, 50 ms) when the firing-rate is increasing sharply, as well as the number of trials needed to get robust estimates, this analysis is not easy to perform and interpret in this context with our data.

5) p7. The attentional enhancement is compared between the pre- and postsaccadic time-window by performing t-tests on the activity in both windows. Is this also done in a single 2-way ANOVA with factors time window (pre/post) and attentional state (attended/unattended)? The statement "a direct comparison between the attentional effects in the two time periods also showed that the attentional effect in the presaccadic period was greater than that in the postsaccadic ..." is a bit ambiguous. What is tested here and how? A significant interaction effect in the ANOVA as mentioned above is expected, but I'm not sure that's what was done.

We used a paired t-test to estimate the significance of the difference between the pre-saccadic and post-saccadic attentional effects. We have updated the sentence (pages 7-8) so that it now reads – “Consistent with these results from the separate significance tests, a direct statistical comparison between the attentional effects in the two time periods also showed that the attentional effect in the pre-saccadic period was greater than that in the post-saccadic period in Experiment 1 (paired t-test; $p < 0.0001$ in both monkeys), and the attentional effect in the post-saccadic period was greater than that in the pre-saccadic period in Experiment 2 (paired t-test; monkey H: $p < 0.0001$, monkey E: $p = 0.001$).”

6) To explain the difference between the present results and previous studies reporting pre-saccadic RF shifts, the authors suggest that there are two separate systems: one that predicts the consequence of an eye movement (including e.g. LIP) and one that reflects that as change in attentional modulation, locked to the saccade (including MT). It seems that an alternative explanation might be that the studies without pre-saccadic shifts, there was a distractor item in the RF that needed to be suppressed, whereas in the studies with pre-saccadic RF shifts there typically was no item in the neuron's RF before the saccade. Or are there reasons to reject this as an alternative explanation?

There is no pre-saccadic “predictive” activity in MT even in situations where there is no stimulus in the RF before the saccade. We have emphasized this in the current submission to make this more explicit. In the Results section (page 9), we say – “Importantly, we did not find any evidence for predictive attention shifts in MT: there was no attentional enhancement of the post-saccadic target representation before saccade offset (black diamonds in Figures 2a,b). This is particularly notable because unlike earlier studies(Ong and Bisley 2011, Yao, Treue et al. 2016), we made sure that there was a stimulus in the RF before the saccade. The presence of this stimulus ensured a stimulus-driven response on which a putative predictive attentional signal could act, and rules out the argument that the apparent absence of a predictive response is simply because the predictive attentional signal does not modulate spontaneous activity in MT. ”. We then reiterate this in the Discussion section (page 16): “We did not find any evidence for attentional enhancement of the post-saccadic target representation before the saccade, even though our experimental design ensured that there was a (distractor) stimulus in the RF before the saccade, and therefore a distractor-driven response before the saccade on which an attentional effect could be seen, if present. This confirms results in two previous studies (Ong and Bisley 2011, Yao, Treue et al. 2016) without a stimulus in the RF before the saccade, where no significant attentional modulation of spontaneous activity was found in MT neurons before the saccade.”

7) There are peaks visible in Fig. 3b, which are not visible in Fig. 3a. Fig. S2 seems to reveal that there are timing differences between saccades in the two attention conditions. The authors may want to point out the cause of the two peaks in Fig. 3b. Furthermore, were there indeed systematic difference in saccadic latency between the attention conditions? What caused them? Can this have influenced the conclusions?

In our previous submission, we split the attend-in and attend-out trials into two groups (with latencies above and below the median latency) based on the median latencies calculated separately for the attend-in and attend-out conditions – thus, the trials included in the short-latency saccade subset in the attend-in condition may have been systematically faster (or slower) than the ones in the attend-out condition. Since the difference plot (in Figure 3) is based on subtracting the activity between the attend-in and attend-out conditions, using different previous choice is problematic (as the Reviewer points out). We have now redone the latency analysis: for each monkey, we pooled the trials from the attend-in and attend-out conditions before calculating the median. We also matched the saccade offset timing distribution (within successive non-overlapping 20 ms time bins) in the attend-in and attend-out conditions for Experiment 1 and 2 (in each monkey) by dropping trials as necessary (“mean-matching”). The peaks following the mean saccade offset time in Figure 3 b and in Supplementary Figure 3 b,d are now also visible in Figure 3a and Supplementary Figure 3a,c (though the small amplitude differences in the peaks may potentially be the result of some additional offset-time dependence or saccade-cue aligned effects on these modulation peaks).

As for the Reviewer's second question, we now state this in the Supplementary Information: "There were often differences in saccadic latency between the attention conditions, due to interactions between the RF location and the saccade target location with respect to the fixation point. These differences were however usually small and unsystematic: for Experiments 1 and 2 in monkey E, and Experiment 2 in monkey H, 90 % of the sessions had mean latency differences (between the two attentional conditions) that lay below 12.5 ms. The mean differences across sessions were also small and not significant: Experiment 2 in monkey H, 1.3 ms shorter (SEM=0.7 ms), Experiment 1 in monkey E, 5.5 ms shorter (SEM=3.7 ms) and Experiment 2 in monkey E, 4.5 ms shorter (SEM=2.3 ms). The only exception was for Experiment 1 in monkey H. Here, the cutoff value was larger: 90 % of the sessions had mean latency differences below 33.5 ms. Also, these differences were clearly systematic: the saccade latency in the attend-in condition was 20.5 ms shorter than that for the attend-out condition (SEM=1.4 ms).

There are two reasons why these latency differences are unlikely to influence our conclusions. First, they were generally small and non-significant, and even in the case of Experiment 1 in Monkey H, the differences were still of the order of 20-30 ms. Second, and more importantly, the attention-shifts in the data are saccade-synchronized, and therefore these latency differences in the timing of the saccade do not affect the data when viewed aligned to saccade-offset (though they do affect the interpretation when aligned to saccade cue-onset). In any case, we redid the analysis in Figure 2, using the reduced mean-matched dataset from Figure 3, where for each session, trials were dropped from each condition in order to match the distribution of latencies (within 20 ms bins). Though only 66 % of trials were now retained (overall), the basic features of the attentional time-course in the 0-100 ms period after saccade offset remain robust (Supplementary Figure 7)."

Minor

1) p12. Ln14-16. I understand the message, but I get lost in this sentence. There may be a word missing (around 'sufficient'), but I think this could be rewritten more clearly.

Yes, we agree that the sentence was very unwieldy. We have now rewritten it on page 12 as – "This indicated that the neural response to a change occurring only 30 ms after saccade offset receives sufficient attentional enhancement that the detection performance for this change is as good as that for changes occurring at later times."

2) I think there's something going wrong with the panel referencing in the legend of Sup fig 2. The legend talks about short and long latency saccades in panel a-d and e-h respectively, but I guess this should be a-b,e-f and c-d,g-h.

We thank the Reviewer for catching this error. We have fixed it.

3) On p. 9, line 3 the reader wonders what these IQRs are and only finds out in the methods that

they are related to bootstrapping. This information should be added when the IQRs are first mentioned.

We have now added the information. Also, we have now amended our bootstrap procedure to make it more conservative (by additionally incorporating across-neuron variability into the procedure) – this changed the IQRs by less than 5 ms. We also realized that our validation procedure for the bootstrap may not be robust when the number of trials or neurons is small, or when there is model uncertainty. Determining a good bootstrap procedure for our analysis (based on the time-course of multi-level spike-train datasets) will require much more statistical investigation. However, because we think it is still a useful indicator, we still mention the IQR from our conservative procedure, but only in connection with the main result on Page 9 (that is based on the largest dataset).

4) Legend Fig. 2, line 15: the curves in c and d are magenta and cyan, not red and blue.

We thank the Reviewer for catching this error. We have fixed it.

5) Is there a relation between the latency of postsaccadic attentional modulation and that at presaccadic stimulus onset? I realize that this is only possible for the neurons that have been recorded in both experiments.

We appreciate the Reviewer's question, but this analysis is not feasible for two reasons: a) unlike response latency, estimation of **attentional** latency within single neurons requires many more trials than we have, given the attentional effect-size and the dependence of latency measures on firing-rate. b) because the cue was presented within the RF, the cue response/attentional effect actually preceded the RDP onset in experiment 1 (see Figure 1). In any case, we have appended a figure (Reviewer Response Figure 2) showing the average PSTHs for the post-saccadic attentional modulation in Experiment 2 (aligned to saccade offset) and the pre-saccadic attentional modulation (aligned to RDP onset) in Experiment 1.

Bahill, A. T., M. R. Clark and L. Stark (1975). "Dynamic overshoot in saccadic eye movements is caused by neurological control signed reversals." Exp Neurol **48**(1): 107-122.

Doron, G. and M. Brecht (2015). "What single-cell stimulation has told us about neural coding." Philos Trans R Soc Lond B Biol Sci **370**(1677): 20140204.

Falkner, A. L., M. E. Goldberg and B. S. Krishna (2013). "Spatial representation and cognitive modulation of response variability in the lateral intraparietal area priority map." J Neurosci **33**(41): 16117-16130.

Nystrom, M., I. Hooge and K. Holmqvist (2013). "Post-saccadic oscillations in eye movement data recorded with pupil-based eye trackers reflect motion of the pupil inside the iris." Vision Res **92**: 59-66.

Ong, W. S. and J. W. Bisley (2011). "A lack of anticipatory remapping of retinotopic receptive fields in the middle temporal area." J Neurosci **31**(29): 10432-10436.

Yao, T., S. Treue and B. S. Krishna (2016). "An Attention-Sensitive Memory Trace in Macaque MT Following Saccadic Eye Movements." PLoS Biol **14**(2): e1002390.

Zenon, A. and R. J. Krauzlis (2012). "Attention deficits without cortical neuronal deficits." Nature **489**(7416): 434-437.

Reviewer Response Figure 1. The time-course of activity around RDP onset in Experiment 1 (a) and saccade-offset in Experiment 2 (b). PSTHs calculated by filtering spike-trains with a 5 ms standard-deviation Gaussian window (stepping every 1 ms). We used a 5 ms filter since the stimulus response latency was a critical variable here.

Reviewer Response Figure 2. The attentional enhancement of the post-saccadic target representation is sufficient to allow perfect attentional decoding (within 50 ms) with a realistic number of independent Poisson neurons. The plot shows the proportion of trials in which a population of independent Poisson neurons representing the target RDP would produce more spikes in total within a 50 ms window when compared to 3 other competing populations (representing the other three RDPs). See text for details.

Reviewer Response Figure 1. The time-course of activity around RDP onset in Experiment 1 (a) and saccade-offset in Experiment 2 (b). PSTHs calculated by filtering spike-trains with a 5 ms standard-deviation Gaussian window (stepping every 1 ms). We used a 5 ms filter since the stimulus response latency was a critical variable here.

Reviewer Response Figure 2. The attentional enhancement of the post-saccadic target representation is sufficient to allow perfect attentional decoding (within 50 ms) with a realistic number of independent Poisson neurons. The plot shows the proportion of trials in which a population of independent Poisson neurons representing the target RDP would produce more spikes in total within a 50 ms window when compared to 3 other competing populations (representing the other three RDPs). See text for details.

Original Supplementary Figure 1. The attention shift remains saccade-synchronized when the firing-rates are matched between the populations in Experiment 1 and 2 in monkey E. Related to Figure 2. Figure format identical to Figure 2, except that only data from monkey E are shown, and the 25 % of highest-firing rate neurons from Experiment 2 were excluded from the sample. The attentional cross-over time now occurred at a median time of 56 ms after saccade offset.

REVIEWERS' COMMENTS:

Reviewer #1 (Remarks to the Author):

The authors have adequately addressed my previous comments.

Reviewer #2 (Remarks to the Author):

Saccade-synchronized, rapid attention shifts in macaque visual cortical area MT
Tao Yao, Stefan Treue, B. Suresh Krishna

-- First revision --

The authors have done an excellent job thoroughly revising the manuscript and replying to the reviewers' comments and questions. The additional analyses and figures on the distribution of target change timing relative to the saccade are a strong addition to the paper. Overall, the more thorough discussion of the results in this revision have also improved the paper.

I remain supportive of publication and trust that the authors can address the following minor comments in their final manuscript version:

- 1) The additional information on the saccade endpoints and corrective saccades are informative. While I appreciate the authors reluctance to speculate about RF sizes, I would like to see a brief statement somewhere in the paper to reiterate their answer to the reviewers, i.e. that they "did not always precisely map the RF size (...) but made sure that the relevant pre- and post-saccadic targets lay inside/outside the receptive field". Even with the slight variability in saccade endpoints."
- 2) In Fig. 1c, the cyan curve exhibits a strong response before time zero and only in the target condition. This suggests a response to the cue in the target conditions which must have fallen in the RF for some cells. Please discuss this (unexpected?) response and include appropriate controls to demonstrate that this was not a confound.
- 3) Supp Info, p. 2 line 4: should this reference perhaps be to Supp Fig. 4?

Response to the Reviewers

Reviewer #1 (Remarks to the Author):

The authors have adequately addressed my previous comments.

We thank the Reviewer for the extensive and useful comments to the earlier version, that enabled us to substantially improve the manuscript.

Reviewer #2 (Remarks to the Author):

Saccade-synchronized, rapid attention shifts in macaque visual cortical area MT

Tao Yao, Stefan Treue, B. Suresh Krishna

-- First revision --

The authors have done an excellent job thoroughly revising the manuscript and replying to the reviewers' comments and questions. The additional analyses and figures on the distribution of target change timing relative to the saccade are a strong addition to the paper. Overall, the more thorough discussion of the results in this revision have also improved the paper.

I remain supportive of publication and trust that the authors can address the following minor comments in their final manuscript version:

We thank the Reviewer for the positive summary.

1) The additional information on the saccade endpoints and corrective saccades are informative. While I appreciate the authors reluctance to speculate about RF sizes, I would like to see a brief statement somewhere in the paper to reiterate their answer to the reviewers, i.e. that they "did not always precisely map the RF size (...) but made sure that the relevant pre- and post-saccadic targets lay inside/outside the receptive field". Even with the slight variability in saccade endpoints"

As the Reviewer suggests, we have now added this text to the Methods section (page 30):

“Given the complex nature of our main task, as well as the difficulties of mapping the edges of the RF, we did not always map the RF size before starting the main task: we only ensured that the post-saccadic stimulus in Experiment 1 lay outside the pre-saccadic RF (and vice-versa for Experiment 2), even with the slight variability in saccadic endpoints. We also ensured that the RF locations shifted commensurately with fixation position (i.e. the RFs were retinotopic).”

2) In Fig. 1c, the cyan curve exhibits a strong response before time zero and only in the target condition. This suggests a response to the cue in the target conditions which must have fallen in the RF for some cells. Please discuss this (unexpected?) response and include appropriate controls to demonstrate that this was not a confound.

This response (that the Reviewer refers to) is the expected response to the presentation of the cue within the RF before the saccade – in other words, the cue appeared at the hand-mapped RF center for *all* cells. The monkey then attended to the RDP at the RF location before the saccade. This sort of cueing is standard practice to elicit attentional effects in the field, and produces attentional effects qualitatively similar to those from symbolic cueing. Since the peri-saccadic dynamics that we focus on occurs around 1 second after the cue disappearance, we do not believe that this confounds our results or interpretation in any manner.

This confusion probably emerged because we introduced some unfortunate errors in the Legend to Figure 1 when we changed the figure colors at some point. We mistakenly used the words “(blue curve)” and “(red curve)” in this sentence - *In Experiment 1, after the saccade, the stimulus in the RF was always a distractor, while before the saccade, either a target (blue curve) or a distractor (red curve) appeared in the neuron’s RF; the situation was reversed for Experiment 2* – and have now removed these words.

Also, we had the words “(blue curve)” when we meant “(cyan curve)” in this sentence - The early response before RDP onset in the attend-in condition (blue curve) is the cue response.

We have now modified the legend to correctly reflect the actual conditions tested. We have also carefully proof-read the manuscript again and have hopefully fixed all remaining errors/typos of this kind.

3) Supp Info, p. 2 line 4: should this reference perhaps be to Supp Fig. 4?

We thank the Reviewer for catching this error: the reference is now correct (and to Supp Fig 4).